# GestureMaster: Graph-based Speech-driven Gesture Generation

Chi Zhou
NetEase Games AI LAB
Hangzhou, Zhejiang, China
zhouchi1@corp.netease.com

Tengyue Bian
NetEase Games AI LAB
Hangzhou, Zhejiang, China
biantengyue@corp.netease.com

Kang Chen
NetEase Games AI LAB
Hangzhou, Zhejiang, China
ckn6763@corp.netease.com

## ABSTRACT

This paper describes the GestureMaster entry to the GENEA (Generation and Evaluation of Non-verbal Behaviour for Embodied Agents) Challenge 2022. Given speech audio and text transcriptions, GestureMaster can automatically generate a high-quality gesture sequence to accompany the input audio and text transcriptions in terms of style and rhythm. GestureMaster system is based on the recent ChoreoMaster publication[12]. ChoreoMaster can generate dance motion given a piece of music. We make some adjustments to ChoreoMaster to suit for the speech-driven gesture generation task. We are pleased to see that among the participating systems, our entry attained the highest median score in the human-likeness evaluation. In the appropriateness evaluation, we ranked first in upper-body study and second in full-body study.

## CCS CONCEPTS

• **Computing methodologies → Procedural animation**.

## KEYWORDS

Gesture Generation, Audio-driven Pose Estimation

### ACM Reference Format:

Chi Zhou, Tengyue Bian, and Kang Chen. 2022. GestureMaster: Graph-based Speech-driven Gesture Generation. In *INTERNATIONAL CONFERENCE ON MULTIMODAL INTERACTION (ICMI '22), November 7–11, 2022, Bengaluru, India.* ACM, New York, NY, USA, 7 pages. https://doi.org/10.1145/3536221.3558063

## 1 INTRODUCTION

Non-verbal-behaviour such as gestures are vital in human communication. Automatically generating high-quality gestures from audio and text transcriptions remains a challenging task. The GENEA Challenge 2022 [21] on speech-driven gesture generation aims to bring together researchers that use different methods for non-verbal-behaviour generation and evaluation.

Recent deep learning-based approaches like StyleGestures[1] have successfully been applied to synthesizing gesture poses. These methods grasp some deeper relationships between audio, text transcriptions and gestures than traditional techniques. However, these methods are limited by the representation power of proposed neural networks. Neural networks characterize data by projecting it into a

low-dimensional latent space, while high-frequency motion details of gestures are considered to be noised and internally ignored. This lowers the quality of generated gestures, causing them to be "dull" and "blurred".

We have developed GestureMaster system. It is adjusted from recent music-to-dance system Choreomaster[12]. Given paired audio, text transcriptions and gestures, we first build a gesture database. This database consists of gesture clips split from training gestures by an automatically split algorithm. Then, we find a style signature by StyleGestures-like network by mapping audio into a desired gesture feature, and a rhythm signature for each clip of audio and gestures. The style signature and the rhythm signature are then incorporated within a graph-based motion synthesis framework. It can generate high-quality gestures with high-level human-likeness and high appropriateness score for the associated held-out speech, in terms of timing or rhythm. To improve the smoothness of gesture transitions, in the graph search, the rotations are interpolated by Slerp of adjacent motion clips.

## 2 RELATED WORK

In this section, we discuss previous work in related areas.

### 2.1 Graph-based Motion Synthesis

Graph-based motion synthesis has long been an important topic in computer animation. Lamouret et al. [15] proposed the first prototype system to synthesize motions by cutting-and-pasting together existing motion clips from the database. Arikan et al. [2], Kovar et al. [13] and Lee et al. [17] formally introduced the concept of graph-based motion synthesis, casting the problem as finding paths in a pre-constructed *motion graph*. Lee et al. [18] proposed *motion fields*. They mapped motion data into a high-dimensional generalization of a vector field. Then they trained a reinforcement learning model to generate responses to user input. Clavet and Büttner [3] proposed *motion matching*, which is a k-Nearest Neighbor search method of searching a large database of animations for the animation which best fits the given context and has been widely used in video games. Holden et al. [11] proposed learned motion matching approach to generate locomotion animation with a neural network regressor.

Recently, Kang et al. [12] proposed ChoreoMaster, a production-ready music-driven dance motion synthesis system. Given a piece of music, ChoreoMaster could build a motion graph and search matched motion clips with a dynamic programming algorithm. Our key idea is directly derived from ChoreoMaster and we adapt this system to gesture generation.

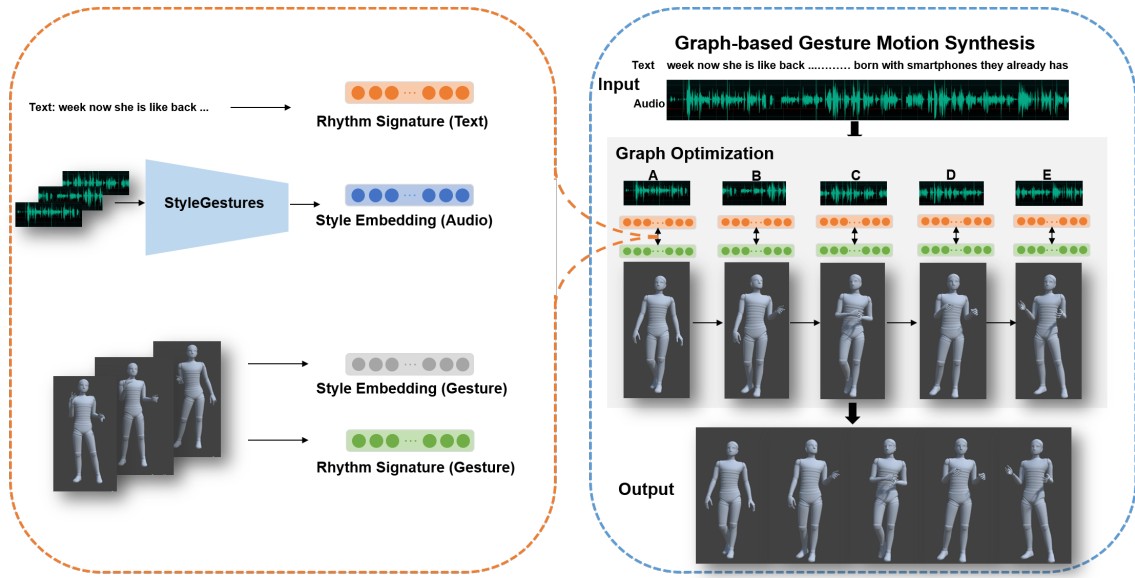

**Figure 1: Overview of our proposed system GestureMaster. Given an input of audio and text transcriptions, we split them into clips. For each audio clip, we calculate its rhythm signature (see Figure 2) and style signature using StyleGestures. Then the graph-based gesture motion synthesis module searchs matched gesture motion nodes from database with lowest cost, in terms of rhythm, style and transition (see equation 2).**

## 2.2 Co-speech Gesture Generation

Deep learning approaches have been widely used in co-speech gesture generation. Hasegawa et al. [10] generated 3D motion sequences using a bi-directional LSTM network given speech audio. Kucherenko et al. [14] incorporated representation learning for the motion to generate smoother gestures, given audio as input only. Yoon et al. [20] trained a LSTM model on TED-talk videos to map text transcriptions to 2D gestures. Alexanderson et al. [1] proposed StyleGestures which was based on normalizing flows to generate gestures with different styles such as hand height and hand velocity.

Ferstl et al. [4] trained a classifer to automatically detect gesture phases. Ferstl et al. [6] analysed the predictability of gesture parameters such as velocity, initial acceleration, size, arm swivel and hand opening. Then Ferstl et al. [7] proposed a database-driven approach. They build a large corpus of co-speech gestures and selected individual gestures based on predicted gesture parameters given speech audio. Yang et al. [19] utilized a graph-based framework to successfully synthesize body motions for social conversations. Recently Habibie et al. [9] proposed a motion matching-based framework for controllable gesture synthesis from speech and the 3D gesture was passed to a conditional GAN to refine gesture sequence. Different from these approaches, GestureMaster add a rhythm embedding module and a style embedding module into searching framework to improve appropriateness and use a global dynamic programming optimization to generate gesture sequences.

## 3 DATA PREPARATION

The dataset provided by the challenge organizer is adapted from Talking With Hands 16.2M[16]. It comes from recording of dyadic interactions between different speakers. Each dyad has been separated into two independent sides with one speaker each. The training dataset inlcudes 293 recordings with an overall length of 18 hours. Each recording consists of audio, text transcripts and gesture motion.

## 3.1 Gesture Segmentation

Gesture phases are a hierarchy of movements that composes or describes gesticulation. Identifying actual gesture phases (rest position, preparation, stroke, hold, retraction/recovery, and partial recovery) is non-trivial and in many cases requires subjective judgment. For example, Ferstl et al. [4, 5] trained a classifer to automatically detect gesture phases. The labelling process of gesture phases cannot be seen as deterministic and 100% accuracy is unlikely. For simplification, we do not consider the type of gesture phases in our system. We just split each gesture motion in training dataset into clip-level gesture clips automatically by time interval of words larger than 0.4 seconds in text transcriptions. These gesture clips can be used to build a motion graph for graph-based optimization in section 4.

The choice of time interval is a trade-off between human likeness and appropriateness. In our experiments, we use two criteria to choose the hyperparameter time interval threshold:

1. The average time of gesture clips should not be too large. Otherwise the total number of gesture clips is small, which decreases the diversity of rhythm of gesture clips in the database and lowers the final appropriateness score.

2. The average time of gesture clips should not be too small. Otherwise given a speech with same duration, more gesture clips

and transitions are needed to generate whole gesture sequences, which lowers the final human likeness score.

The choice of time interval threshold is also related to the speech rate of different speakers. In Talking With Hands 16.2M dataset, based on these two criteria, we find setting time interval threshold equal to 0.4 seconds lead to best generated gestures.

We manually remove gesture clips with low quality such as motions with jitter or wrong rotations. For motion capture without finger animation, we simply search and transfer the rotations from finger motion capture with lowest position distance. Then these gesture clips are semi-automatically annotated with rhythm signatures and style signatures for graph-based motion synthesis. Finally We mirror the gesture clips and build a gesture database with more than 6000 clips, range from 1 seconds to more than 10 seconds, determined by the length of each clip.

## 3.2 Rhythm Embedding

The term rhythm is often expressed in terms of beat. Beat corresponds to pulses of sound in audio, while gesture motion beat corresponds to pausing or sharp truning of gesture movements. The proposed rhythm signature consists of 32 bits in our system (see Figure 2). In each rhythm signature, bits denote the presence of beats (1 : present, 0 : not present) which correspond to the evenly-spaced beats indicated by the time signature. For rhythm signature of audio and text transcriptions, bits denote the presence of words. For rhythm signature of gesture motion, bits denote the presence of pausing, sharp turning or stroke gesture. Obviously, a time of silence will result in a rhythm signature in which all bits are zeros. The distance between two rhythm signatures can be defined using Hamming distance, the number of bit positions in which the two bit patterns differ. Lower hamming distance indicates a better match of rhythm between audio and gesture motion.

For audio, rhythm signature is annotated automatically using word-level timing information in text transcriptions (see Figure 2). For each audio clip, we use the present time of each word in text transcriptions and label the bits as 1 automatically.

For gestures, rhythm signature is annotated automatically using speed curve of two hands (see Figure 3). We compute the max speed curve of two hands for each gesture clip and record the time of local minima. We suppose the local minima denotes the start time of pausing, sharp turning or stroke gesture and label the bits as 1 automatically. For each gesture clip, the attached rhythm signature is a vector consists of 32 bits. After automatically labeling of rhythm signature of gesture clips, we manually correct the rhythm signature of gesture clips which the speed curves fail to represent the presence of pausing, sharp turning or stroke gesture.

## 3.3 Style Embedding

StyleGestures is a probabilistic model which could generate gestures with different style, such as gesture speed, radius and height. We splice these features together as a style signature. We calculate mean speed, mean radius and mean height of each gesture clip in database offline. And we adopt StyleGestures as a backbone of style embedding network and train the model on training dataset. In synthesis period, we could feed audio into StyleGestures to generate

desired gestures and style signature for graph-based optimization (see Figure 1).

## 4 SYSTEM OVERVIEW

In this section, we discuss the pipeline of proposed system (see Figure 1), including motion graph construction and graph-based optimization. We explain how the rhythm embedding module and style embedding module are incorporated into our graph-based motion synthesis framework.

### 4.1 Motion Graph Construction

A motion graph is a directed graph where each node denotes a motion clip in the database while each edge depicts the cost of transition between two adjacent nodes.

In our system, each node in our motion graph corresponds to a gesture clip. In our motion graph, the edge transition cost $T(D_p, D_q)$ between two nodes $D_p$ and $D_q$ is defined as:

$$T(D_p, D_q) = \lambda_1 T_p + \lambda_2 T_r \tag{1}$$

Where $T_p, T_r$ is summed distance of positions, rotations between joints in transitional frames of two adjacent nodes, respectively. $\lambda_1$ and $\lambda_2$ are the corresponding weights. An edge is created in the graph if the transition cost between adjacent nodes is below a threshold $\delta_T$. A higher $\delta_T$ results in more edges in the graph but may also cause artifacts as bad transition edges may also be included in the graph.

We build motion graph for upper body and lower body separately. For upper body motion graph, a style signature and a rhythm signature are also attached to each graph node.

### 4.2 Graph-based Optimization

In the graph-based framework, each synthesized motion corresponds to a path in the motion graph. In our system, gesture generation can be viewd as finding optimal paths. Given audio and text transcriptions, we first split transcriptions into several clips with time interval threshold 0.4 seconds and we obtain an audio sequence $M = \{M_i | i = 1, \ldots, n\}$, where $M_i$ represents clip $i$ of input audio. Then we calculate the rhythm signature $R_{M_i}$ of $M_i$ (see Figure 2). The goal of our system is to assign a gesture motion node $D_i$ in the motion graph to each $M_i$ and to minimize the following cost:

$$C = \lambda_3 \Sigma_{i=1}^{n} C_d(i) + \lambda_4 \Sigma_{i=1}^{n-1} C_t(i, i+1) \tag{2}$$

where $C_d, C_t$ are the data term and transition term, respectively. $\lambda_3, \lambda_4$ are the corresponding weights.

*Data term.* $C_d(i)$ is the sum of rhythm signature and style signature matching cost between audio clip $M_i$ and motion node $D_i$:

$$C_d(i) = \lambda_5 G_z(Z_{M_i}, Z_{D_i}) + \lambda_6 G_r(R_{M_i}, R_{D_i}) \tag{3}$$

where $G_z, G_r$ are style signature L2 distance and rhythm signature Hamming distance between audio and gestures. Style signature of audio clip $Z_{M_i}$ is calculated using StyleGestures while style signature of gesture clip $Z_{D_i}$ is calculated offline (see Section 3.3). $\lambda_5, \lambda_6$ are weights. Specifically, for optimization of lower body, we set $\lambda_5 = \lambda_6 = 0$ and data term $C_d(i) = 0$.

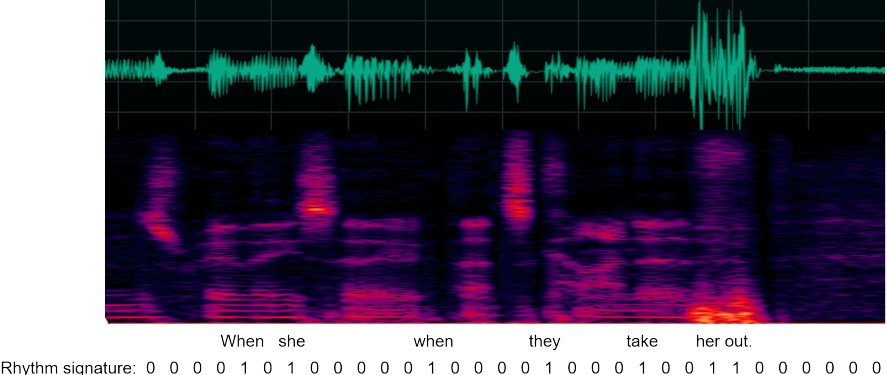

Figure 2: Rhythm signature of audio examples. Bits denote the presence of words (1 : present, 0 : not present).

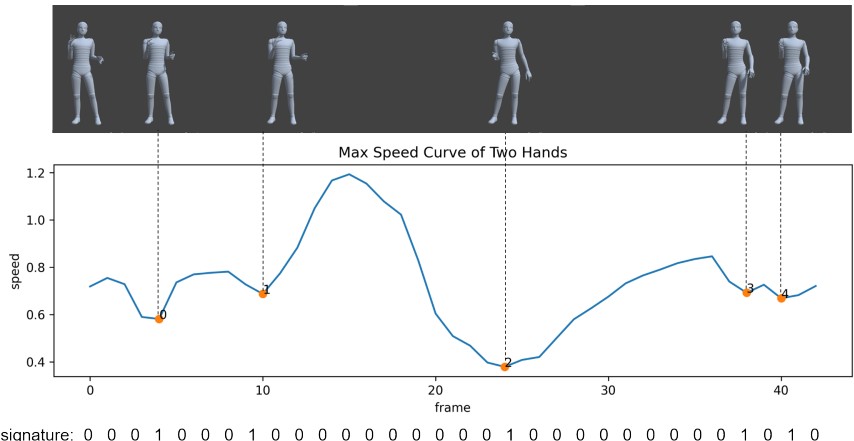

Figure 3: Rhythm signature of gestures examples. Bits denote the presence of pausing, sharp turning or stroke gesture (1 : present, 0 : not present).

*Transition term.* $C_t$ ensures a smooth transition between adjacent motion clips in the synthesized motion.

$$C_t(i, i+1) = T(D_i, D_{i+1}) \qquad (4)$$

The optimal gesture motion sequences are synthesized using a dynamic programming algorithm[8]. We handle the transitions in the graph search using Slerp interpolation between two adjacent gesture clips. After generating upper body motion and lower body motion separately, we blend the two motions to create a full body motion. We smooth all synthesized gestures using Savitzky-Golay filter. For arm, hand and head joints, the length of the filter window and the order of the polynomial are 7 and 2 while for other joints are 7 and 1, respectively.

## 5 EVALUATION

For upper body graph, we set the hyperparameters to: $\lambda_1, \ldots \lambda_6 = 0.7, 0.3, 1.0, 3.0, 1.0, 0.1, \delta_T = 8, \zeta = 10000$. For lower body graph, we set the hyperparameters to: $\lambda_1, \ldots \lambda_6 = 0.7, 0.3, 0.0, 0.0, 0.0, 0.0, \delta_T =$

$8, \zeta = 0$. Our gesture synthesis system is tested on a desktop with a 3.70GHz i7-8700K CPU, 32GB RAM and a GTX 3070 GPU.

Study participants were recruited through a crowdsourcing platform. Participants were required to reside in a set of six English-speaking countries, specifically UK, IE, USA, CAN, AUS, and NZ, and participants were required to have English as their first language. Each study incorporated attention checks per person, to make sure that participants were paying attention to the task and remove insincere test-takers.

The evaluation of the submitted gesture motion will likely consider two aspects such as its perceived **human-likeness**, without accounting for the speech and its **appropriateness** for the associated held-out speech, in terms of timing and semantic content. Study participants were recruited through the crowdsourcing platform Prolific. The groundtruth natural motion was labelled **FNA** in the fullbody study and **UNA** in the upper-body study. Our condition ID in the upper-body evaluation was **USQ** and our condition ID in the full-body evaluation was **FSA**. The evaluations also included two baseline systems, one based on text-input only [20], and one based on audio-input only [14].

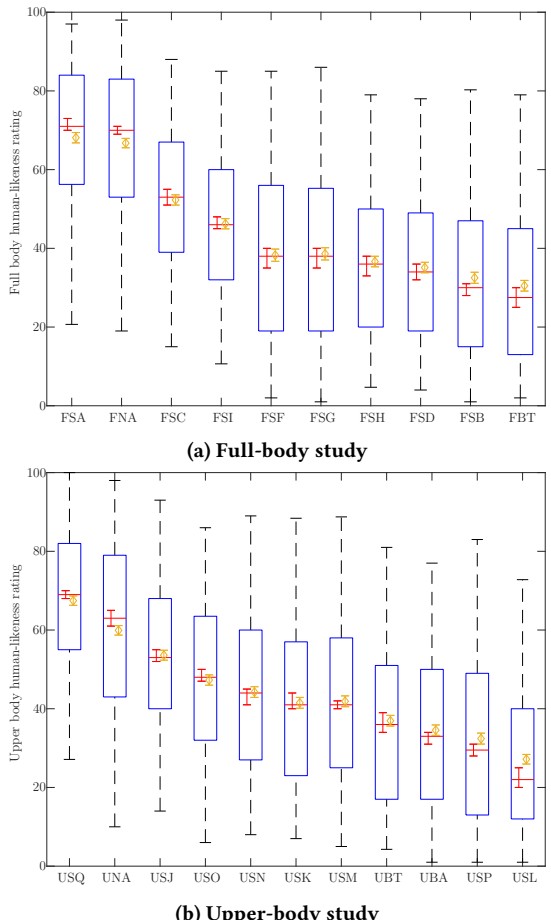

(a) Full-body study

(b) Upper-body study

Figure 4: Box plots visualising the ratings distribution in the two studies. Red bars are the median ratings (each with a 0.05 confidence interval); yellow diamonds are mean ratings (also with a 0.05 confidence interval). Box edges are at 25 and 75 percentiles, while whiskers cover 95% of all ratings for each condition. Conditions are ordered descending by sample median for each tier.

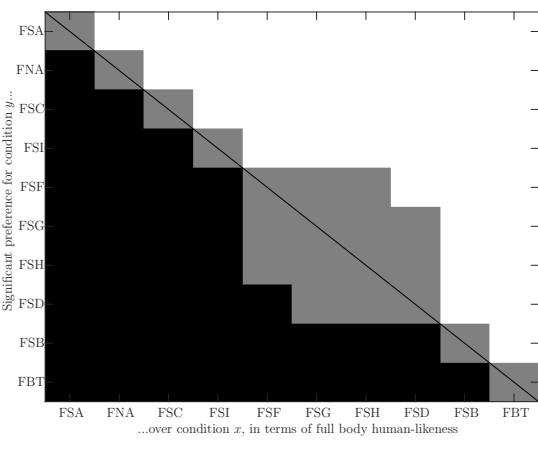

(a) Full-body study

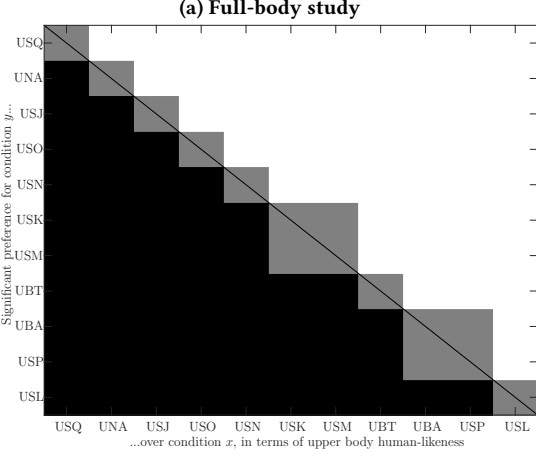

(b) Upper-body study

Figure 5: Significance of pairwise differences between conditions. White means that the condition listed on the $x$-axis rated significantly above the condition on the $y$-axis, black means the opposite ($y$ rated below $x$), and grey means no statistically significant difference at the level $\alpha = 0.05$ after Holm-Bonferroni correction. Conditions are listed in the same order as in Figure 4, which is different for each of the two studies.

## 5.1 Human Likeness Study

121 participants successfully passed the attention checks and completed the full-body human-likeness study, while 150 participants successfully passed the attention checks and completed the upper-body human-likeness study. In human likeness study, study participants were asked "How human-like does the gesture motion appear?" then gave their ratings in response to this question on a scale from 0 (worst) to 100 (best). GestureMaster (**FSA**, **USQ**) ranked first and even above the groundtruth motion from the motion-capture recordings in both full-body and upper-body tiers. Bar plots and significance comparisons are shown in Figure 4 and Firure 5. Summary statistics (sample median and sample mean) for the ratings of all conditions in each of the two studies are shown in Table 1. The human likeness study shows that GestureMaster could generate natural gestures.

## 5.2 Appropriateness Study

247 participants successfully passed the attention checks and completed the full-body appropriateness, while 304 participants successfully passed the attention checks and completed the upper-body appropriateness study. In appropriateness study, participants were given pair of videos – both from the same condition and thus having the same motion quality, but one matched to the speech and the other mismatched, coming from unrelated speech. Participants were then asked to pick the one video from the pair that best matched the speech. GestureMaster (**FSA**, **USQ**) ranked first in upper-body tier and second in full-body tier. Bar plots are shown in Figure 6.

Benefit from the matching rhythm signature of audio and gestures, the appropriateness for the associated held-out speech perform well, in terms of timing.

| | Human-likeness | | Appropriateness | | | |
|---|---|---|---|---|---|---|
| | | | Number of responses | | | Percent matched |
| ID | Median | Mean | Match. | Equal | Mismatch. | (splitting ties) |
| FNA(GT) | 70 ∈ [69, 71] | 66.7 ± 1.2 | 590 | 138 | 163 | 74.0 ∈ [70.9, 76.9] |
| FBT | 27.5 ∈ [25, 30] | 30.5 ± 1.4 | 278 | 362 | 250 | 51.6 ∈ [48.2, 55.0] |
| FSB | 30 ∈ [28, 31] | 32.5 ± 1.5 | 397 | 163 | 330 | 53.8 ∈ [50.4, 57.1] |
| FSC | 53 ∈ [51, 55] | 52.3 ± 1.4 | 347 | 237 | 295 | 53.0 ∈ [49.5, 56.3] |
| FSD | 34 ∈ [32, 36] | 35.1 ± 1.4 | 329 | 256 | 302 | 51.5 ∈ [48.1, 54.9] |
| FSF | 38 ∈ [35, 40] | 38.3 ± 1.6 | 388 | 130 | 359 | 51.7 ∈ [48.2, 55.1] |
| FSG | 38 ∈ [35, 40] | 38.6 ± 1.6 | 406 | 184 | 319 | 54.8 ∈ [51.4, 58.1] |
| FSH | 36 ∈ [33, 38] | 36.6 ± 1.4 | 445 | 166 | 262 | **60.5 ∈ [57.1, 63.8]** |
| FSI | 46 ∈ [45, 48] | 46.2 ± 1.3 | 403 | 178 | 312 | 55.1 ∈ [51.7, 58.4] |
| **FSA(Ours)** | **71 ∈ [70, 73]** | **68.1 ± 1.4** | 393 | 216 | 269 | 57.1 ∈ [53.7, 60.4] |

(a) **Full-body study**

| | Human-likeness | | Appropriateness | | | |
|---|---|---|---|---|---|---|
| | | | Number of responses | | | Percent matched |
| ID | Median | Mean | Match. | Equal | Mismatch. | (splitting ties) |
| UNA(GT) | 63 ∈ [61, 65] | 59.9 ± 1.3 | 691 | 107 | 189 | 75.4 ∈ [72.5, 78.1] |
| UBA | 33 ∈ [31, 34] | 34.6 ± 1.4 | 424 | 264 | 303 | 56.1 ∈ [52.9, 59.3] |
| UBT | 36 ∈ [34, 39] | 37.0 ± 1.4 | 341 | 367 | 287 | 52.7 ∈ [49.5, 55.9] |
| USJ | 53 ∈ [52, 55] | 53.6 ± 1.3 | 461 | 164 | 365 | 54.8 ∈ [51.6, 58.0] |
| USK | 41 ∈ [40, 44] | 41.5 ± 1.4 | 454 | 185 | 353 | 55.1 ∈ [51.9, 58.3] |
| USL | 22 ∈ [20, 25] | 27.2 ± 1.3 | 282 | 548 | 159 | 56.2 ∈ [53.0, 59.4] |
| USM | 41 ∈ [40, 42] | 41.9 ± 1.4 | 503 | 175 | 328 | 58.7 ∈ [55.5, 61.8] |
| USN | 44 ∈ [41, 45] | 44.2 ± 1.4 | 503 | 175 | 328 | 58.7 ∈ [55.5, 61.8] |
| USO | 48 ∈ [47, 50] | 47.3 ± 1.4 | 439 | 209 | 335 | 55.3 ∈ [52.1, 58.5] |
| USP | 29.5 ∈ [28, 31] | 32.4 ± 1.4 | 440 | 180 | 376 | 53.2 ∈ [50.0, 56.4] |
| **USQ(Ours)** | **69 ∈ [68, 70]** | **67.5 ± 1.2** | 504 | 182 | 310 | **59.7 ∈ [56.6, 62.9]** |

(b) **Upper-body study**

**Table 1: Summary statistics of user-study ratings from all user studies, with confidence intervals at the level $\alpha = 0.05$. "Percent matched" identifies how often participants preferred matched over mismatched motion in terms of appropriateness.**

## 6 CONCLUSION

We have proposed GestureMaster, a graph-based gestures synthesis system. We build a gesture database including more than 6000 gesture clips with style signature and rhythm signature. Given audio and text transcriptions, a graph-based optimization is adopted to generate high-quality gesture motion. The evaluation results demonstrate that GestureMaster can synthesize gestures with high human-likeness score as well as high appropriateness score for associated speech in terms of rhythm.

There is a gap between GestureMaster and ground truth motion in the appropriateness study. In future research, a better rhythm embedding module could be used for better rhythm matching. Semantic content could also be considered to improve appropriateness. Because of imbalanced data, we do not evaluate the appropriateness for the individual gesticulation style of the indicated test speaker in each clip. Howerer, GestureMaster could simply generate gestures for each speaker by building different graphs for different speakers, indicating the potential of GestureMaster to generate individual-related gestures.

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
