# OpenReview forum: "GestureMaster: Graph-based Speech-driven Gesture Generation"
_ACM.org/ICMI/2022/Workshop/GENEA — GENEA Challenge & Workshop 2022 Mainproceeding_

### Official Review · Reviewer_3ZhP · 2022-08-09
**Review of GestureMaster entry to the GENEA Challenge 2022**

**Rating:** 8
**Confidence:** 4

**Review:**

Few minor typos. E.g. Section 1p3 “. This database consists of gesture phases split from training gesture motion by an automatically split algorithm. T” and Section 2 p2 last sentence accidentally capitalized “We.” There are a few others throughout the paper and although it does not impact the comprehensibility of the paper, the readability of the paper would benefit from a thorough proofread.

2.1 It is unclear to me from the paper whether beats were recovered manually or algorithmically, and if algorithmically what algorithm was used – off the shelf, previously published, or a part of this work.

3 – very comprehensive discussion of motion graph and optimization algorithms. Could use some introduction to the motion graph concept before this as as it reads now it jumps directly into some gritty details without first describing how it will be used in an overarching algorithm. Providing a bit of pseudo-code or a high-level algorithmic description before jumping in and then sign-posting to sections in that generative algorithm would be very helpful.

4 – good explanation of results and evaluations. In Table 1 highlighting the models described in this paper would improve readability and immediate understanding of this paper’s contributions and place in the rankings. It would also be helpful to understand why this model performed so well. The authors provide some insight into why these could be expected to perform well for human-likeness, but why so well with semantic appropriateness? As I don’t see any place semantic content is included in the generation, why did this model perform so well?

Clearly very strong algorithm but needs deeper discussion into why the authors think it performed so well. I think this paper should be accepted provided those additions.

---

### Official Review · Reviewer_CSSm · 2022-08-09
**This paper presents an alteration to ChoreoMaster for gestures, which involves finding gesture phases and using a rhythm embedding.**

**Rating:** 6
**Confidence:** 5

**Review:**

Nice idea to adapt Choreomaster to gestures, and a promising approach taken using gesture phases, rhythm embeddings and graph search.

A few issues that should be addressed to improve the paper:

No related work section, and authors are missing a lot of citations to back-up their statements and observations about gestures. The authors are working in the area of gesture synthesis, but have made some decisions about gesture phasing and labeling words as beats, which need to be justified and compared to prior work in the area. Particularly other papers that use phase-based approach to human motion (e.g., Phase-functioned neural networks for character control, Holden et al.) and gestures (e.g., Adversarial gesture generation with realistic gesture phasing, Ferstl et al.).

Identifying actual gesture phases (rest position, preparation, stroke, hold, retraction/recovery, and partial recovery) is a complex task and the authors claim to be splitting into gesture phases but they are actually just splitting by time intervals of words larger than 0.4 seconds in the text transcripts, which is not clear to me how these are classified as gesture phases. More details on gestures phases and how this is done are required. Or authors should change the name to something more appropriate. Please also refer to the gesture phasing in Ferstl et al. and how this method compares. Please also discuss how your method compares to others that use gesture phasing and graph-based search (see ExpressGesture: Expressive gesture generation from speech through database matching).
How did you handle the transitions in the graph search?

-Lots of typos in the text, please run a spell check.
-One that should be fixed for future experiments is 'How human-likeness does the gesture motion appear?' which should be 'How human-like does the gesture motion appear?'
-A justification for the unusal scale of 0 to 100 should be made also, 1-5 or 1-7 point Likert scales are more common for these types of questions.
-More details on the statistical tests you performed are needed.
-Graphs in Figure 5 are not standard and require more explanation.
-How many participants took part in the experiment?
-Did you check on Prolific if they were native English speakers and were able to assess the appropriateness of the speech to the gestures?
-Did you include some attention checks to be sure they were completing the experiment correctly, which is common for crowd-sourced experiments?

---

### Official Review · Reviewer_MmEw · 2022-08-11
**Interesting direction of gesture motion maps**

**Rating:** 7
**Confidence:** 5

**Review:**

### Summary

The authors propose a gesture generation model that is conditioned on speech audio and text transcriptions. The key idea here is derived from ChoreMaster, which relies on the constructed motion graph to generate relevant gestures given speech and language inputs. This generation model is optimized in a dynamic programming setting, which makes it computationally efficient. This model is evaluated using subjective evaluation metrics to measure naturalness and appropriateness of generation.

### Strengths
- This method would generate very natural gestures as the generated gestures are taken directly from the ground truth recordings.
- The optimization of this approach is computationally efficient as it can be solved using a dynamic program

### Weaknesses

- My main concern is with the data preparation where each phase is defined as the gesture where a word was spoken more than 0.4 seconds.
  - Why is this definition of phase useful? Some experiments in support of this design choice would have been useful
  - A common definition of gesture phase involves looking at the fine grained structure of a gesture. There are pre-gesture, main gesture, post-gesture and neutral phases which is quite different from the definition in this paper. It might be good to clarify the differences in the definition of gesture phase here.

Some suggestions:
- In eq 3, one of the cost function is the distances between rhythm signatures between audio and gesture. Are the rhythm signatures between audio and gestures similar? A small experiment to verify this fact could add a lot of value to this paper.
- The naturalness of the generation is high which is not a surprise as the sequence of body motions are taken from the ground truth. But the appropriateness is still lesser that 0.5 indicating that the gestures, although natural looking, might be random. The readers might be interested to hear more about appropriateness as well.
- The literature review on co-speech gesture generation is practically missing from the paper
- I would also cite motion matching papers which have the same basic ideas on generating body motion by choosing the relevant collection of snippets from ground truth data.
- It would increase the readability of the paper if the baselines in this paper were defined or cited

---

### Decision · Program_Chairs · 2022-08-11

**Decision:**

Accept (Main proceeding)

**Comment:**

Congratulations, your paper was accepted to the GENEA Challenge 2022 and is going to be published in the main ICMI Proceedings, given that the camera-ready version is provided on time.

Reviewers agree that the paper describes the systems well, but has several issues which need to be addressed. Most notably, splitting into gesture phases requires more details and clarification and related work needs to be properly reviewed. See below for the full reviews.

We suggest that the authors carefully consider the feedback received from the reviewers and use it to improve their manuscript for the camera-ready submission.